# Evaluation of Sierra Leone's Elimination of Mother-to-Child Transmission of HIV program, 2024: The need for a Life Stages approach to triple elimination

Mariama Mustapha[1], Ibrahim Turay[2], Finda P. Pessima[2], Francis K. Tamba[2], Semion Saffa-Turay[3], Gerald Younge[2], Veronica L. Deen[2], Eshetu Kebede Tabor [2,4], Tagoola Abner[2], Ginika Egesimba[2], Ibrahim Franklyn Kamara[4], Basil Uguge[5], Godswill Agada[5], Silas Quaye[6], Tony T. Ao[6], Andrew Abutu[7], Matilda N. Kamara[8], Osman Sankoh[9,10], Ernest Kenu[11], Isaac Ahemesah[3], Sartie M. Kenneh[8], Austin Demby[8], Sulaiman Lakoh[8,12]*

**1** Research Unit, KYM Consultancy Limited, Freetown, Sierra Leone, **2** National AIDS Control Program, Ministry of Health, Government of Sierra Leone, Freetown, Sierra Leone, **3** The Joint United Nations Agency against HIV/AIDS, UNAIDS Country Office in Sierra Leone, Freetown, Sierra Leone, **4** World Health Organization, WHO Country Office in Sierra Leone, Freetown, Sierra Leone, **5** Global Reach Program, Jhpiego Sierra Leone, Freetown, Sierra Leone, **6** US CDC, Division of Global HIV and TB, Accra, Ghana, **7** US CDC, Division of Global HIV and TB, Freetown, Sierra Leone, **8** Ministry of Health, Government of Sierra Leone, Freetown, Sierra Leone, **9** Centre for Health Research and Training, University of Management and Technology, Freetown, Sierra Leone, **10** School of Community Health Sciences, Njala University, Bo Campus, Bo, Sierra Leone, **11** School of Public Health, University of Ghana, Accra, Ghana, **12** College of Medicine and Allied Health Sciences, University of Sierra Leone, Freetown, Sierra Leone

* lakoh2009@gmail.com

## Abstract

The elimination of Mother-to-Child Transmission (eMTCT) Cascade Analysis is a key intervention to understand the effectiveness and gaps in national HIV prevention program for pregnant women and their infants. This comprehensive evaluation aimed to assess the step-by-step processes of eMTCT services, from the entry points at antenatal care (ANC) to outcomes for HIV-exposed infants (HEIs) and propose actionable recommendations using the life stages approach to improve maternal and child health outcomes in Sierra Leone. We used a retrospective cohort study to understand the entry of HIV-positive pregnant women into the eMTCT program and the outcome of their infants. The assessment was conducted in June 2024 across 118 selected health facilities nationwide. Data was analysed using Microsoft Excel to assess the steps from entry into the eMTCT program to HIV testing outcomes and delivery process for infants, including final infant/child outcome for HIV-positive pregnant and breastfeeding women. Of the 80,264 pregnant women attending their first ANC visit between January and December 2022, 74,401(92.7%) received HIV tests. Of the 2,083 newly diagnosed HIV-positive pregnant women, 1,612 (77.4%) were initiated on antiretroviral therapy (ART) at their first ANC visit, highlighting a critical gap

**Data availability statement:** The data supporting this article is available from: https://figshare.com/articles/dataset/Rapid_eMTCT_casecade_analysis_in_Sierra_Leone/30473198?file=59139068. Identifier: 10.6084/m9.figshare.30473198.

**Funding:** This work was supported by the Global Fund to fight HIV, Tuberculosis and Malaria to SK and the Joint United Nations Agency against HIV/AIDS to IA. The funders had no role in study design, data collection and analysis, decision to publish, or preparation of the manuscript.

**Competing interests:** The authors have declared that no competing interests exist.

where 22.6% missed early treatment initiation. Only 753 (36.1%) HIV-positive women had documented facility-based delivery, indicating significant drop-offs in retention and continuity of care. Among 761 HEIs, 671 (88.2%) received ART for prophylaxis within 72 hours of birth; however, only 466 (61.2%) underwent their first virological test within 6–8 weeks of birth, and 176 (24.1%) received a second virological test 12 weeks after cessation of breastfeeding. While Sierra Leone's eMTCT program has made substantial progress in ANC attendance and HIV testing, critical gaps persist in ART initiation, facility delivery of HIV-positive women and retention in care. Adopting a life stage approach is critical to address these gaps through targeted interventions, improved data quality, and enhanced follow-up care.

## Introduction

HIV remains a major global public health issue, having claimed an estimated 42.3 million lives to date [1]. Transmission is ongoing in many countries [2]. An estimated 39.9 million people were living with HIV globally at the end of 2023, 65% of whom are in the WHO African Region [2]. In 2023, an estimated 630,000 people died from HIV-related causes, and 1.3 million people became newly infected with HIV [1]. In Sierra Leone, 77,000 people were living with HIV and 1,800 died from AIDS-related illness in 2023 [3].

Mother-to-child transmission (MTCT) of HIV is a public health challenge, particularly in regions with high HIV prevalence [4]. The transmission of HIV from an HIV-positive mother to her child can occur during pregnancy, labor, delivery, or breastfeeding [5]. Without intervention, the risk of MTCT ranges from 15% to 45%. However, effective prevention measures can reduce this risk to less than 2% in non-breastfeeding mothers or less than 5% in breastfeeding populations [4,5].

Globally, about 90% of children living with HIV acquire their infection from their mothers during pregnancy, birth, and breastfeeding [6]. In Sierra Leone, the program for the elimination of mother-to-child transmission (eMTCT) started in 2004 and has been rolled out to achieve national coverage, which currently stands at only 87% as of December 2023 [7,8]. EMTCT is a cascade of services involving HIV testing, provision of prompt and efficacious treatment, safe delivery, appropriate follow-up of exposed infants, optimal infant feeding, and lifelong treatment and care for mothers living with HIV to prevent transmission of the infection to their children [4]. HIV testing among pregnant women happens in the context of antenatal care (ANC) and during labor and delivery. Sierra Leone has achieved universal ANC coverage at 78.2%, and 98% of women received ANC from a skilled provider at least once in 2019 [9,10]. This high ANC skilled attendance, however, has not resulted in optimal coverage of HIV testing for pregnant women and follow-up care for HIV-positive mothers and their infants [8].

Although Sierra Leone has shown an increase in the percentage of HIV-positive women who were put on ART – from 51% in 2009 to 87% in 2023 – the country is still far from meeting the eMTCT target of 95% [8,11]. The sub-optimal ART coverage

can be attributed to the non-universal HIV testing coverage for all pregnant women, as well as the low coverage of comprehensive eMTCT sites. Furthermore, the country is not doing well in early infant diagnosis (EID). In 2023, the country reported only 9% of infant HIV testing within two months of birth [8]. Data on infant HIV testing at 9 and 18 months are not available, which presents a challenge in determining the final MTCT outcomes and accounting for MTCT during breastfeeding.

Conducting a rapid eMTCT cascade analysis can provide critical insights into the effectiveness of the existing eMTCT program in Sierra Leone, measure progress, identify gaps in service delivery and data management, and inform targeted interventions to improve eMTCT service delivery and health care quality and outcomes for mothers and infants.

This comprehensive evaluation aimed to assess the step-by-step processes of eMTCT services, from the entry points at ANC to the final outcomes for HIV-exposed infants (HEI) and generate actionable recommendations for improving maternal and child health outcomes. Furthermore, the cascade analysis informed the establishment of a functional ANC sentinel surveillance system in the country.

## Materials and methods

### Study design

This retrospective cohort study was implemented to assess the entry of an HIV-positive pregnant woman into the eMTCT program and the outcome of the infants at the point of exit. The eMTCT cascade analysis and ANC site assessment was conducted in June 2024 across selected health facilities nationwide.

### Study setting

Sierra Leone is a West African country, bordered by Guinea to the north and east, Liberia to the southeast, and the Atlantic Ocean to the west and southwest. The country has five administrative regions and 16 health districts [10,12]. Sierra Leone has an estimated population of 8. 7 million with an annual growth rate of 2.13% [12]. It has an estimated infant mortality rate of 70.5 per 1,000 live births and an under-five mortality rate of 98.1 per 1,000 live births [12,13]. The healthcare system in Sierra Leone is under-resourced. It faces numerous challenges, including limited access to healthcare services, a high burden of infectious diseases such as malaria, and infant deaths [14,15]. With funding from the Global Fund, The President's Emergency Plan for Aids Relief (PEPFAR), and other Health Development Partners, eMTCT activities have been scaled up in the past few years in Sierra Leone. There are currently about 960 sites in Sierra Leone that provide eMTCT services but little is known about the uptake of services, acceptance of HIV testing, and the status of syphilis and hepatitis B testing.

### Study population

The study included all pregnant women of reproductive age (15–49) who accessed ANC services from January to December 2022. HIV-exposed infants (HEI) of pregnant women in this cohort were included and followed up to determine their final HIV status.

### Sampling procedure

We used the UNAIDS/WHO methodology to select a convenient sample of 118 health facilities that reported data on eMTCT variables from January to December 2022 for the cascade analysis and site assessment [16]. The selection of these facilities reflects different geographical locations (urban vs. rural) and HIV burden. The health facilities were distributed across the 16 Sierra Leone districts, including the existing ANC sentinel survey sites. They represent 43% of ANC testing volume and 77% of HIV positivity. Increasing the ANC testing volume did not yield better coverage regarding positivity.

We used a cut-off of health facilities that reported 6 HIV-positive pregnant and breastfeeding women in one year. This arbitrary figure of 6 was decided based on the country's routine program data, which indicates a 5–6% HIV positivity rate. Additionally, the cascade analysis tracked variables such as HIV-positive pregnant and breastfeeding women, including exposed infants, to assess service quality and outcomes. Having adequate quantity in each assessed health facility was critical to the eMTCT cascade analysis.

## Data collection

Data was collected using a locally adapted collection tool (S1 Table). Data was abstracted from routine facility registers using three pre-tested data collection tools on June 20, 2024 in accordance with national data protection guidelines and ethical standards for secondary data analysis. The authors do not have access to information that could identify individual participants during and after data collection. A hybrid approach was utilized for data collection. Quantitative data elements that were electronically recorded were collected using the Open Data Kit (ODK) software. Using the appropriate registers, we recorded all pregnant women who had their first ANC visit (ANC 1) and were tested for HIV during the specified period. We then created a line list of all clients who tested HIV positive during their first ANC visit to track their progress through delivery and postnatal care. This involved documenting the services the women and their children received, including ART, post-exposure prophylaxis, EID services, and rapid diagnostic test (RDT). Trained teams assessed health facilities' readiness and availability, including collecting and analyzing data using standard tools. The teams collaborated with HIV counsellors, and ANC, labour, and post-natal staff at the respective health facilities to complete the assessment. They verified and documented all existing data collection tools and source documents listed in the data abstraction tools. Data entered from the field was synced in real time except during work in areas with poor internet coverage. The central dashboard and data repository systems had access levels to guarantee data security. Paper abstracted data was cross-checked at the end of each day for accuracy and completeness.

## Data analysis

The data collected was downloaded to a Microsoft Excel workbook and cleaned by employing processes to query errors, including outliers. The data was then coded, and a pre-developed Excel eMTCT cascade analysis tool was used to analyze data. Data entered into the Excel eMTCT cascade analysis tool was analyzed in steps from entry into the eMTCT program to HIV testing outcomes and delivery of infants, including final infant/child outcomes for HIV-positive pregnant and breastfeeding women.

## Ethical consideration

The Sierra Leone Ethics and Scientific Review Committee of the Ministry of Health, Government of Sierra Leone, approved the application for non-research determination prior to the start of the study in accordance with the relevant guidelines and regulations and declaration of Helsinki. The approval number is 014/06/2024. Routine secondary data (including HIV test results) were abstracted under a waiver of informed consent. The research team did not have access to information that could identify individual participants during or after data collection. All the authors adhered to ethical standards during the research and publication.

## Results

### ANC site assessment of health facilities with eMTCT program

A total of 118 ANC sites in health facilities was assessed. Of these, 37.3% (44/118) were in the Western Area. On monthly average ANC, 59.3% (70/118) of the facilities had less than 50 ANC attendees. All the 118 facilities perform triple testing (HIV/HBV/Syphilis) using the opt-in approach. Onsite eMTCT HIV testing is nearly evenly split between urban 52.5%

(62/118) and rural areas 47.5% (56/118), with a slight urban majority. Syphilis testing is routinely done all year, with a slight urban majority of 52.5% (62/118) (Table 1).

## Logistics availability and payment for testing services

Out of the 118 health facilities assessed, 35.6% (42/118) facilities had the capability to perform triple testing onsite. Most [81.0% (34/42)] of the facilities were located in the urban setting. Regarding payment for eMTCT services, 11.3% (7/62) of the facilities in the urban setting charged for eMTCT services. However, none of the facilities in the rural setting received payment from clients for eMTCT services. HIV test kit stockouts are more frequent in rural areas, 54.8% (40/73) compared to urban settings. In 2023, 59.7% (43/72) of the health facilities with stockout of HIV test kits experienced at least two instances of stockout within the year (Table 2).

## Maternal eMTCT cascade analysis

Between January and December 2022, we assessed 80,264 pregnant women who attended the first ANC visit. Of these, 80,264 ANC attendees, 92.7% (74,401/80,264) received HIV testing during their first ANC visit. Of all the newly tested

**Table 1. Characteristics of health facilities with eMTCT Program, June 2024, Sierra Leone.**

| eMTCT Program Characteristics | Residence | | |
|---|---|---|---|
| | Urban (N = 62) n (%) | Rural (N = 56) n (%) | Total (N = 118) n (%) |
| **Region** | | | |
| East | 7 (36.8) | 12 (63.2) | 19 (16.1) |
| North | 5 (45.4) | 6 (54.5) | 11 (9.3) |
| Northwest | 4 (16.7) | 20 (83.3) | 24 (20.3) |
| South | 13 (65.0) | 7 (35.0) | 20 (16.9) |
| West | 33 (75.0) | 11 (25.0) | 44 (37.3) |
| **ANC registrants** | | | |
| < 50 registrants | 30 (42.9) | 40 (57.1) | 70 (59.3) |
| ≥ 50 registrants | 32 (66.7) | 16 (33.3) | 48 (40.7) |
| **PMTCT Testing** | | | |
| Onsite | 62 (52.5) | 56 (47.5) | 118 (100.0) |
| **PMTCT Testing approach** | | | |
| Opt-in | 62 (52.5) | 56 (47.5) | 118 (100.0) |
| **Syphilis testing** | | | |
| Syphilis testing done routinely all year | 62 (52.5) | 56 (47.5) | 118 (100.0) |
| **Nature of syphilis Testing** | | | |
| Onsite | 62 (53.0) | 55 (47.0) | 117 (99.2) |
| Offsite | 0 (0.0) | 1(1.8) | 1 (0.8) |
| **HBV testing part of routine ANC** | | | |
| HBV testing done routinely all year | 24 (72.7) | 9 (27.3) | 33 (28.0) |
| HBV testing done routinely only during surveillance | 3 (100.0) | 0 (0.0) | 3 (2.5) |
| HBV testing not done routinely | 35 (42.7) | 47 (57.3) | 82 (69.5) |
| **Nature of HBV Testing** | | | |
| Onsite | 34 (81.0) | 8 (19.0) | 42 (35.6) |
| Offsite | 28 (36.8) | 48 (63.2) | 76 (64.4) |

*ANC-Antenatal Clinic, eMTCT-elimination of mother-to-child transmission, HBV-Hepatitis B Virus, PMTCT-Prevention of Mother-to-Child Transmission*

**Table 2. Logistics for testing services in Health facilities, June 2024, Sierra Leone.**

| Characteristics | Residence | | |
|---|---|---|---|
| | Urban (N = 62) n (%) | Rural (N = 56) n (%) | Total (N = 118) n (%) |
| **Payment for eMTCT services** | | | |
| Yes | 7 (11.3) | 0 (0.0) | 7 (5.9) |
| No | 55 (88.7) | 56 (100.0) | 111 (94.1) |
| **HIV Test Kits stockout** | | | |
| Yes | 33 (53.2) | 40 (71.4) | 73 (61.9) |
| No | 29 (46.8) | 16 (28.6) | 45 (38.1) |
| **Instances of stockout** | | | |
| Once | 14 (42.4) | 15 (37.5) | 29 (39.7) |
| Twice | 19 (57.6) | 25 (62.5) | 44 (60.3) |
| **Triple testing** | | | |
| Yes | 34 (54.8) | 8 (14.3) | 42 (35.6) |
| No | 28 (45.2) | 48 (85.7) | 76 (64.4) |

*eMTCT-elimination of mother-to-child transmission*

HIV-positive women, 77.4% (1,612/2,083) were initiated on ART at their first ANC visit, and of those that tested positive, 36.1% (753/2,083) had documented facility-based delivery (Fig 1).

## Gaps in the maternal eMTCT cascade

The number clients not tested for HIV on their first ANC visit was 5,863 (7.3%). The Eastern region had the highest proportion, 49.5% (2,904/5,863), of ANC attendance not tested for HIV on their first visit, while the Northern region had the lowest proportion, 4.3% (253/2,896).

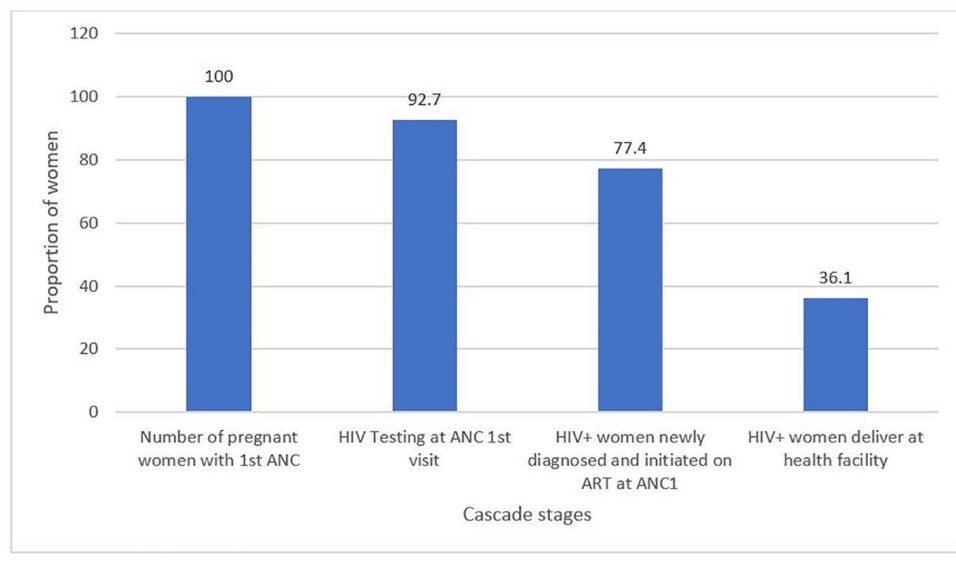

*ANC-antenatal care      ART-antiretroviral therapy, HIV-Human Immunodeficiency Virus*

**Fig 1. Maternal eMTCT cascade of health facilities, June 2024.**

Also, out of the 2,083 HIV positive women, 22.6% (471) were not initiated on ART with the Eastern region contributing the largest proportion (31.8%, 150/471). Out of 2, 083 HIV-positive pregnant women, 63.9% (1330/2083) did not deliver at the health facility. The Western Area had the highest proportion 56.6% (753/1330) of HIV+ women who did not deliver in the health facility (Table 3).

### Infant eMTCT cascade analysis

Of all the women who tested positive, 36.1% (753/2,083) delivered 761 exposed infants in the health facilities. A high proportion of 88.2% (671/761) of HEIs received NVP prophylaxis within 72 hours after birth and 61.2% (466/761) had their first virological test within 6-8 weeks of birth. Only 6.4% (30/466) of the exposed infants tested positive for HIV after the first virological test, and 24.1% (176/731) of HEIs had a second virological test done 12 weeks after cessation of breast-feeding. Of the 176 exposed infants with the second test, 16.5% (29/176) tested positive (Fig 2).

### Gaps in the infant eMTCT cascade

Out of 761 exposed infants, 11.8% (90/761) were not provided ARV prophylaxis within 72 hours of birth, 38.8% (295/761) did not receive their first virological test within 6-8 weeks of birth, and 75.9% (555/731) were without second virology test. Out of the 106 HIV-positive infants, 6.6% (7/106) were not linked to care (Table 4).

## Discussion

In this first, large scale, eMTCT cascade analysis in Sierra Leone, we highlighted four key findings on the maternal and infant HIV care cascades. First, only 35.6% of the facilities, mostly in urban areas, could perform triple testing. Second, 92.7% of pregnant women received an HIV test at their first antenatal visit and 77.4% of those with a positive result were receiving ART, but only 36.1% had documented deliveries in health facilities. Third, 61.2% of the positive newborns underwent the first virological test within 6–8 weeks after birth, 6.4% of which were positive, and 24.1% of the newborns underwent the second test 12 weeks after stopping breastfeeding, 16.5% of which were positive. Finally, NVP prophylaxis was missed in 11.8% of HEIs.

With the scale-up of eMTCT services across the country in the past several years, an MTCT rate of 16.5% is disappointing as Sierra Leone is far from achieving the global target of 5% in breastfeeding populations and 2% in non-breastfeeding populations [5]. There is need for a concerted and proactive approach among all stakeholders for strategic investment to identify and place all HIV positive women of reproductive age and pregnant women on sustained ART and improve on infant HIV services in line with the current national and international guidelines. While this finding is similar to the UNAIDS reported MTCT rate of 19% for Sierra Leone, it is in contrast with MTCT rate in Kenya (8.9%) and that of

**Table 3. Gaps in maternal eMTCT cascade of health facilities, June 2024, Sierra Leone.**

| Region | ANC Registrants not tested for HIV (N = 5,863) n (%) | HIV+ clients not on ART (N = 471) n (%) | HIV+ clients who deliver outside a health facility (N = 1,330) n (%) |
|---|---|---|---|
| East | 2,904 (49.5) | 150 (31.8) | 176 (13.2) |
| North | 253 (4.3) | 22 (4.7) | 56 (4.2) |
| Northwest | 1,014 (17.3) | 120 (25.5) | 183 (13.8) |
| West | 1,692 (28.9) | 139 (29.5) | 753 (56.6) |
| South | 0(0.0) | 40 (8.5) | 162 (12.2) |

*ANC-antenatal clinic, ART-antiretroviral therapy, HIV-Human immunodeficiency virus*

**Global Public Health** PLOS

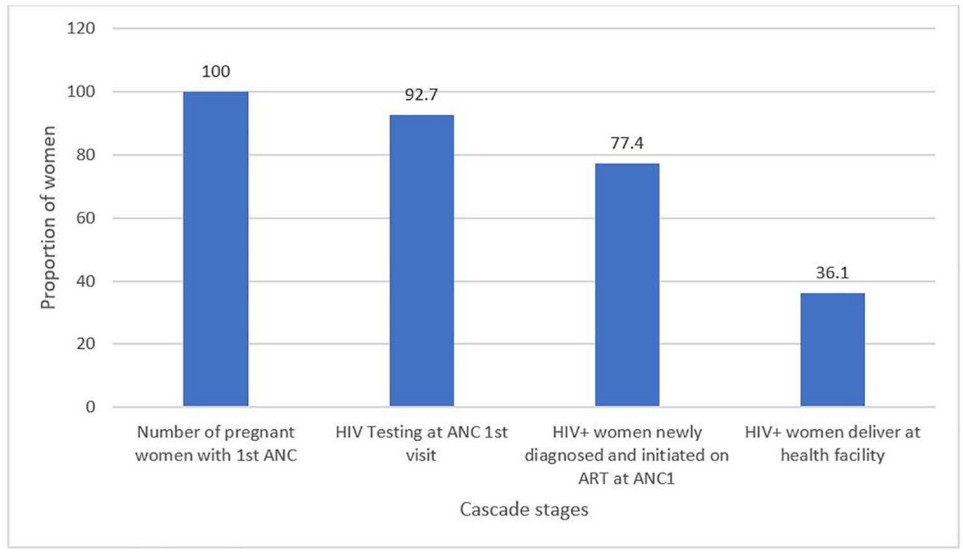

ANC-antenatal care     ART-antiretroviral therapy, HIV-Human Immunodeficiency Virus

**Fig 2. Infant cascade MTCT cascade of health facilities, June 2024.**

**Table 4. Gaps in infant eMTCT cascade of health facilities, June 2024, Sierra Leone.**

| | HIV-Exposed Infants not Initiated on ARV (N=90) n (%) | HIV-Exposed Infants without 1st Virological Test (N=295) n (%) | HIV+ not linked to Care (N=7) n (%) | No 2nd virology test performed (N=555) n (%) | Did Not Receive Test Results (N=66) n (%) |
|---|---|---|---|---|---|
| East | 20 (22.2) | 140 (47.5) | 0 | 103 (18.6) | 0 |
| North | 0 | 5 (1.7) | 1 (14.3) | 21 (3.8) | 1 (1.5) |
| Northwest | 27 (30.0) | 21 (7.1) | 0 | 80 (14.4) | 7 (10.6) |
| West | 39 (43.3) | 91 (30.8) | 6 (85.7) | 300 (54.5) | 57 (86.4) |
| South | 4 (4.4) | 38 (12.9) | 0 | 51 (9.2) | 1 (1.5) |

ARV-antiretroviral

Botswana (2.2%), a country that is on the path for elimination of vertical transmission of HIV [17,18]. Across the globe, 22 countries have been validated by WHO for the elimination of mother-to-child transmission of HIV, or certified on the path to elimination, including Denmark in Europe, along with Bostwana and Namibia in sub-Saharan Africa [19]. WHO awarded Namibia, a high HIV prevalent country, the "silver tier" status for progress on reducing hepatitis B and "bronze tier" for progress on HIV owing to its concerted strategy to curb the transmission of HBV, HIV and syphilis [20]. Namibia applied service integration, using a person-centred approach to improve health outcomes for mothers and children [20].

To eliminate mother-to-child transmission of HIV, women and their infants must be enrolled and retained in the eMTCT continuum of care from ANC through delivery and 18–24 months post-partum. However, many women in our study failed to enroll in care despite several opportunities and many others dropped out or are lost to follow-up in the eMTCT care continuum. The reason why pregnant women drop out of treatment is unclear but may be due to lack of follow-up services for HIV pregnant women or poor health literacy among the HIV positive women.

In March 2024, WHO published the triple elimination framework to encourage countries to address the gaps in vertical transmission of HIV, hepatitis B, and syphilis, including testing [4]. Our study shows that most health facilities in Sierra Leone do not have the capacity to implement the triple elimination, underscoring the need for training, mentoring, and supportive supervision. The Ministry of Health and its partners have begun implementing triple elimination using a life stage approach to ensure that the health needs of our children are addressed throughout their life stage approach. We will strengthen the elimination of these three diseases in line with Sierra Leone's framework of holistic care to HEIs using the country's life stage approach to prevent future vulnerabilities [21].

Despite the progress in HIV testing of pregnant women in Sierra Leone, the 93% of the ANC attendees tested for HIV between January and December 2022 was lower than the 96.3% reported in Ghana and 98.5% reported in Tanzania [22,23]. ANC registration is the first point of contact that allows attendees to be recruited into the cascade based on their test results. This implies that the 7% untested ANC attendees in this study were not recruited into the cascade, denying them the benefits of knowing their HIV status and all the measures to improve their health and eliminate HIV transmission to their babies. The reasons for this gap are unclear but could be investigated in future research. However, to achieve a 100% testing rate for ANC attendees, it is essential to intensify education and individual-level counselling and strengthen HIV surveillance among pregnant women. One of the established approaches is the use of mentor mothers in following up pregnant women and HEIs until the final infant outcome [24]. We will explore the use of mentor mothers to assign cohorts of pregnant women for consistent follow-up of the final infant outcome.

Less than 80% of pregnant women who tested positive for HIV had documented ART initiation at their first ANC visit. Although this is not unique to Sierra Leone, it reflects a worrying trend in the cascade since more than 20% of those who tested positive still left the cascade. The lower rate of documented ART initiation among pregnant women could be explained by challenges with linkage to treatment, poor psychosocial counselling and verticalization of HIV testing. Without ART, the viral load in an HIV-positive pregnant woman remains high, significantly increasing the risk of transmitting the virus to the baby in utero, thereby thwarting the efforts made in the country's attempt to eMTCT [4]. These women are also at higher risk for pregnancy-related complications such as preterm labor, low birth weight, and stillbirth. Historically, Tanzania reported similar ART uptake (82%) among women who tested positive for HIV in 2017 [23]. In Kenya, the rate of ART uptake among eligible pregnant women reported in 2012 is much lower than we reported (only 40% in Naivasha and 27% in Gilgil) [25]. However, it is difficult to make a direct comparison of findings from these studies with our results due to the time laps, underscoring the importance of strengthening counselling and linkage to treatment, care, and support services in Sierra Leone.

While Sierra Leone had over 83% of deliveries in health facilities, the proportion of documented facility-based delivery in our study is unacceptably low at 36%. Clients who deliver outside of health facilities usually miss out on maternal and infant interventions, including zidovudine (AZT) and NVP prophylaxis for HEI [26], thereby increasing the risk of MTCT of HIV [27]. The low rate of facility deliveries may be due to stigma, which leads women to be delivered in facilities where they were not tested for HIV. We recommend using a biometric system and unique identifiers to track pregnancies, no matter where they deliver, so their children can benefit from NVP and AZT prophylaxis. Furthermore, Sierra Leone is implementing community-led monitoring across several districts to gather quantitative and qualitative data about HIV services [28]. This program will be expanded and tailored to all the districts to reduce stigma and other barriers to the low hospital delivery rate among pregnant women with HIV in Sierra Leone.

Along the infant care cascade, about 88% of HEIs received NVP prophylaxis within 72 hours after birth, implying that more than 10% of infants are without ARV prophylaxis. NVP is given to the infant immediately after birth to protect against the virus during this high-risk period. Failure to give the drug increases the risk of transmission in the infant [26]. The number of HEI that missed NVP prophylaxis in our study is lower than reported in Mulago National Referral Hospital in Uganda in 2024 (11.8% vs. 21.9%) [29].

Despite the high rate of initial prophylaxis, 61.2% of the HEIs underwent their first virological test within 6–8 weeks of birth, revealing gaps in early diagnostic follow-up. Moreover, 6% of HEIs tested positive for HIV. This is more than 3.6%

reported in Nigeria, indicating potential transmission during pregnancy or delivery [30]. Furthermore, less than 25% of the infants received a second virological test 12 weeks after the cessation of breastfeeding, with 16.5% showing HIV transmission at this stage. This highlights the need for continued vigilance and repeated testing to ensure early detection and intervention [31]. Mobile phone calls reportedly improve maternal and infant adherence to eMTCT services in western Kenya, and this service will be strengthened in Sierra Leone to address gaps in eMTCT [32]. To ensure these measures are as effective as they should be, the Ministry of Health and its partners will address frequent commodity stock-outs and strengthen laboratory systems. Despite improvements in eMTCT programs, gaps in follow-up testing and treatment adherence continue to impact HIV transmission rates among infants [33]. Studies in Zimbabwe have also shown that while initial NVP prophylaxis uptake is high, follow-up testing rates drop significantly, mirroring the trends observed in this study [34]. Future longitudinal studies to track the long-term outcomes of HIV-exposed infants, as well as implementation studies on ART uptake in pregnant women and ARV prophylaxis in HEIs, will help to further understand and address the gaps in the Sierra Leone eMTCT cascade.

Our study has several strengths. It is the first large-scale nationwide study conducted to assess the eMTCT cascade in Siera Leone. Therefore, the findings can be generalized. Second, the data collection tools were validated, thereby refining complex questions and ensuring uniform data collection. Healthcare professionals with expertise in reading patients' notes and understanding the different terminologies collected the data. This ensures efficient data collection data collection. Finally, we followed the strengthening reporting of observational study (STROBE) guidelines to report our findings.

Although this assessment provides a detailed analysis of the country's performance on key eMTCT indicators, it has some limitations. This dataset is based on the records from routine services that may sometimes be incomplete. Also, since this data does not include sociodemographic information on clients, it is difficult to determine the association between attrition from the cascade and patient characteristics.

## Conclusion

While Sierra Leone's eMTCT program has made substantial progress, particularly in the early stages of the cascade, such as ANC attendance and HIV testing, there are critical gaps in ART initiation, delivery of HIV-positive women in health facilities, and retention in care of HEIs. Our findings call for radical measures for health facilities to test every pregnant woman for HIV, HBV, and syphilis, and to link every pregnant woman who tests positive with treatment, care, and support services. Second, we recommend that community actors (such as mentor mothers) follow up on all pregnant women living with HIV until they have a documented delivery in a medical facility. Finally, all HEI patients should receive ARV prophylaxis according to national guidelines and undergo nucleic acid amplification testing within 6–8 weeks after birth. A life stage approach is essential to achieving these recommendations through targeted interventions, improved data quality, and comprehensive follow-up towards achieving the eMTCT goals and improving health outcomes for mothers and their children in Sierra Leone.

## Supporting information

**S1 Table. Rapid EMTCT Cascade Analysis Tool and Data Sources.**
(DOCX)

## Acknowledgments

We appreciate the support provided by the administrations of the health facilities where the studies were conducted. We are grateful to the data collectors.

## Author contributions

**Conceptualization:** Gerald Younge, Veronica L Deen, Eshetu Kebede Tabor, Tagoola Abner, Basil Uguge, Godswill Agada, Silas Quaye, Tony T Ao, Ernest Kenu, Isaac Ahemesah, Sartie Kenneh, Austin Demby, Sulaiman Lakoh.

**Data curation:** Mariama Mustapha, Ibrahim Turay, Francis K Tamba, Osman Sankoh, Ernest Kenu.

**Formal analysis:** Mariama Mustapha, Ibrahim Turay, Finda P. Pessima, Francis K Tamba, Semion Saffa-Turay, Tagoola Abner, Osman Sankoh, Ernest Kenu, Isaac Ahemesah.

**Funding acquisition:** Semion Saffa-Turay, Isaac Ahemesah, Sartie Kenneh, Austin Demby.

**Methodology:** Mariama Mustapha, Ibrahim Turay, Finda P. Pessima, Francis K Tamba, Semion Saffa-Turay, Tagoola Abner, Ernest Kenu, Isaac Ahemesah, Sulaiman Lakoh.

**Resources:** Semion Saffa-Turay, Isaac Ahemesah, Sartie Kenneh, Austin Demby, Sulaiman Lakoh.

**Supervision:** Semion Saffa-Turay, Isaac Ahemesah, Sartie Kenneh, Austin Demby, Sulaiman Lakoh.

**Validation:** Mariama Mustapha, Francis K Tamba, Ernest Kenu.

**Writing – original draft:** Mariama Mustapha, Ginika Egesimba, Andrew Abutu, Osman Sankoh, Ernest Kenu, Sulaiman Lakoh.

**Writing – review & editing:** Mariama Mustapha, Ginika Egesimba, Ibrahim Franklyn Kamara, Andrew Abutu, Matilda N Kamara, Osman Sankoh, Ernest Kenu, Sulaiman Lakoh.

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
