## [Decision Letter · Decision Letter 0]

6 Jan 2026

PGPH-D-25-03332

Evaluation of Sierra Leone’s Elimination of Mother-to-child Transmission of HIV Program, 2024: The Need for a Life Course Approach to Triple Elimination

Dear Dr. Lakoh,

Thank you for submitting your manuscript to PLOS Global Public Health. After careful consideration, we feel that it has merit but does not fully meet PLOS Global Public Health’s publication criteria as it currently stands. Therefore, we invite you to submit a revised version of the manuscript that addresses the points raised during the review process.

The manuscript has been evaluated by two reviewers, and their comments are available below.

The reviewers have raised a number of major concerns. They feel the manuscript should clearly outline definitions and the study objectives in the introduction, the methodology would benefit from a clearer ethics and consent statement, and the discussion could be expanded to discuss the wider context of the results and discuss successful interventions.

Could you please carefully revise the manuscript to address all comments raised?

We look forward to receiving your revised manuscript.

Kind regards,

Jennifer Tucker, PhD

Staff Editor

Journal Requirements:

Additional Editor Comments (if provided):

Reviewers' comments:

Reviewer's Responses to Questions

**Comments to the Author**

1. Does this manuscript meet PLOS Global Public Health’s publication criteria? Is the manuscript technically sound, and do the data support the conclusions? The manuscript must describe methodologically and ethically rigorous research with conclusions that are appropriately drawn based on the data presented.? Is the manuscript technically sound, and do the data support the conclusions? The manuscript must describe methodologically and ethically rigorous research with conclusions that are appropriately drawn based on the data presented.

Reviewer #1: Yes

Reviewer #2: Yes

2. Has the statistical analysis been performed appropriately and rigorously?

Reviewer #1: No

Reviewer #2: Yes

3. Have the authors made all data underlying the findings in their manuscript fully available (please refer to the Data Availability Statement at the start of the manuscript PDF file)?

The PLOS Data policy requires authors to make all data underlying the findings described in their manuscript fully available without restriction, with rare exception. The data should be provided as part of the manuscript or its supporting information, or deposited to a public repository. For example, in addition to summary statistics, the data points behind means, medians and variance measures should be available. If there are restrictions on publicly sharing data—e.g. participant privacy or use of data from a third party—those must be specified.requires authors to make all data underlying the findings described in their manuscript fully available without restriction, with rare exception. The data should be provided as part of the manuscript or its supporting information, or deposited to a public repository. For example, in addition to summary statistics, the data points behind means, medians and variance measures should be available. If there are restrictions on publicly sharing data—e.g. participant privacy or use of data from a third party—those must be specified.

Reviewer #1: Yes

Reviewer #2: Yes

4. Is the manuscript presented in an intelligible fashion and written in standard English?

Reviewer #1: No

Reviewer #2: Yes

Reviewer #1: General Comments:

The manuscript provides a comprehensive evaluation of Sierra Leone’s Elimination of Mother-to-Child Transmission of HIV (eMTCT) Program. The data presented are crucial for understanding the current state of HIV prevention efforts in the country, and the analysis highlights both progress and significant gaps. While the study successfully outlines key findings, there are areas that require further elaboration and refinement to enhance clarity and impact.

Detailed Comments:

1. Clarity and Structure:

o The introduction effectively sets the context but could be further improved by clearly defining terms such as eMTCT early on. Additionally, the objectives of the study should be explicitly stated as separate bullet points to improve readability. (Paragraphs 63-75)

2. Methodology (Paragraphs 103-171):

o The methodology section is generally well-structured; however, it lacks details on the ethical considerations taken during data collection, particularly regarding consent for secondary data. Including a brief section on ethical compliance and how participant confidentiality was ensured would strengthen this section.

3. Results Presentation (Paragraphs 174-225):

o The results are well-documented; however, the tables could be visually enhanced for better comprehension. Consider using color coding or differing formats to differentiate categories. Furthermore, the gaps identified in the results (e.g., ART initiation, facility-based deliveries) should be discussed in more detail, exploring the underlying causes.

4. Discussion and Context (Paragraphs 230-310):

o While the discussion addresses the implications of the findings, it could benefit from additional comparative analysis with other countries. Incorporating specific examples of successful interventions from similar contexts could provide valuable insights for stakeholders in Sierra Leone. Additionally, expanding on systemic barriers to care identified in the results would enrich this section.

5. Conclusion and Recommendations (Paragraphs 332-337) :

o The conclusion succinctly summarizes findings but lacks specific, actionable recommendations tailored for various stakeholders. Include a clear call to action, emphasizing the role of different entities, such as government agencies, healthcare providers, and community organizations, in implementing changes based on the findings.

6. Data Integrity and Analysis(see Paragraphs 161-170) :

o The use of descriptive statistics is appropriate, but more detailed statistical analyses (e.g., confidence intervals, significance tests) would lend credibility to the findings. Consider addressing potential biases in data collection and how they might affect the results.

7. Future Research Directions (End of Discussion) :

o The manuscript would benefit from a section outlining future research needs. This could include recommendations for longitudinal studies to track long-term outcomes for HIV-exposed infants or assessments of specific interventions aimed at improving ART uptake.

Additional Considerations:

• Ethics and Dual Publication:

o There are no apparent concerns about dual publication based on the content reviewed. However, it is important to confirm that the authors have adhered to ethical standards during their research and publication processes. Be transparent about any potential conflicts of interest and ensure all data sources are adequately cited.

• Formatting and Writing Style:

o There are minor grammatical issues and inconsistencies in formatting that should be addressed to improve the overall professionalism of the manuscript. A thorough proofreading session is recommended before resubmission.

Reviewer #2: Barring minor correction, the article is well written. It is crisp and concise. It identifies the gaps in the program well. The program requires strengthening at various levels to achieve the goal of elimination of mother to child transmission of HIV.

**Do you want your identity to be public for this peer review?** For information about this choice, including consent withdrawal, please see our Privacy Policy..

Reviewer #1: No

Reviewer #2: **Yes:** Dr Mamatha Murad LalaDr Mamatha Murad LalaDr Mamatha Murad LalaDr Mamatha Murad Lala

---

## [Editor Report · Decision Letter 1]

24 Mar 2026

Evaluation of Sierra Leone’s Elimination of Mother-to-child Transmission of HIV Program, 2024: The Need for a Life Course Approach to Triple Elimination

PGPH-D-25-03332R1

Dear Dr. Lakoh,

We are pleased to inform you that your manuscript 'Evaluation of Sierra Leone’s Elimination of Mother-to-child Transmission of HIV Program, 2024: The Need for a Life Course Approach to Triple Elimination' has been provisionally accepted for publication in PLOS Global Public Health.

Best regards,

Tanmay Bagade, Ph.D., MS (O&G), MPH, MHM

Academic Editor